# Charging and ultralong phosphorescence of lanthanide facilitated organic complex

Waygen Thor [1], Yue Wu[1], Lei Wang[1], Yonghong Zhang [1,2], Peter A. Tanner [1✉] & Ka-Leung Wong[1✉]

Emission from the triplet state of an organo-lanthanide complex is observed only when the energy transfer to the lanthanide ion is absent. The triplet state lifetime under cryogenic conditions for organo-lanthanide compounds usually ranges up to tens of milliseconds. The compound **LaL1(TTA)₃** reported herein exhibits 77 K phosphorescence observable by the naked eye for up to 30 s. Optical spectroscopy, density functional theory (DFT) and time-dependent DFT techniques have been applied to investigate the photophysical processes of this compound. In particular, on-off continuous irradiation cycles reveal a charging behaviour of the emission which is associated with triplet-triplet absorption because it shows a shorter rise lifetime than the corresponding decay lifetime and it varies with illumination intensity. The discovery of the behaviour of this compound provides insight into important photophysical processes of the triplet state of organo-lanthanide systems and may open new fields of application such as data encryption, anti-counterfeiting and temperature switching.

[1] Department of Chemistry, Hong Kong Baptist University, Waterloo Road, Kowloon Tong, Hong Kong SAR. [2] State Key Laboratory of Chemistry and Utilization of Carbon Based Energy Resources, Key Laboratory of Oil and Gas Fine Chemicals, Ministry of Education & Xinjiang Uygur Autonomous Region, Urumqi Key Laboratory of Green Catalysis and Synthesis Technology, College of Chemistry, Xinjiang University, Urumqi 830046 Xinjiang, PR China. ✉email: peter.a.tanner@gmail.com; klwong@hkbu.edu.hk

anthanide ions (Ln$^{3+}$) exhibit sharp and specific emission wavelengths but have weak absorption[1]. Light-harvesting techniques therefore employ coordination to organic molecules[2], or doping into strongly-absorbing inorganic chromophore systems[3]. Organic complexes of lanthanide ions comprise a burgeoning research area with a wide spectrum of applications: optical and magnetic resonance imaging[4,5], contactless thermometry[6], drug delivery[7], and organic light-emitting diodes[8], for example. In optical applications, the properties of the triplet state (T$_1$: Fig. 1a) are of utmost importance. Phosphorescence can be switched on in the absence of ligand-to-metal ion energy transfer, for instance, upon coordination of the organic molecule to a La$^{3+}$, Gd$^{3+}$, or Lu$^{3+}$ ion[9]. The energy transfer pathway from an organic antenna to the lanthanide ion usually follows the sensitization from the lowest singlet state, S$_1$, to T$_1$ and then to Ln$^{3+}$[10,11]. Intersystem crossing from S$_1$ to T$_1$ is spin-forbidden but the rate becomes faster upon coordination to heavy metal, such as a lanthanide ion[12–14]. The total T$_1$ to S$_0$ rate is slow at 77 K upon coordination with a diamagnetic lanthanide ion (La$^{3+}$ or Lu$^{3+}$), giving phosphorescence emission lifetimes in the range of up to hundreds of milliseconds[9,15,16]. However, this is an order of magnitude shorter than for the lanthanide compound reported herein with phosphorescence observable up to 30 s with the naked eye.

In recent years, lanthanide ions are popular as radiative centers for persistent luminescence[13,17]. Although both can deliver long-lived luminescence, persistent luminescence and phosphorescence follow different mechanisms (Fig. 1a, b) and may be distinguished by their decay kinetics. Decay from a triplet state usually exhibits a first-order exponential decay law (Fig. 1c) whereas persistent luminescence decay follows a hyperbolic law (Fig. 1d) or a power law[18,19]. The compound reported herein emits a long afterglow and a charging profile, but its exponential decay curve is typical of phosphorescence processes.

In this work, we describe the long-lived phosphorescent organo-lanthanide compound. Under cryogenic conditions using laser excitation, the green luminescence of **LaL1(TTA)$_3$** ([[(4-(4,6-bis(3,5-dimethyl-1H-pyrazol-1-yl)−1,3,5-triazin-2-yl)-N,N-diethylaniline)] tris[4,4,4-trifluoro-1-(2-thienyl)-1,3-butanedionato]) lanthanum (III)) is observable for up to 30 s at 77 K as illustrated (Fig. 1f). Besides this slow T$_1$ to S$_0$ rate, the kinetics profile demonstrates a peculiar charging profile that is unknown for organo-lanthanide complexes. Such charging characteristics are apparent in some published figures of room temperature phosphorescent organic compounds[14,20], but were not commented upon, so that the mechanism is not explained or revealed. Uncovering **LaL1(TTA)$_3$** not only provides insights into its phosphorescence mechanism but also leads to a new dimension of understanding and application for lanthanide complexes. The ultra-long phosphorescence lifetime occurring in the cryogenic state means that this compound can be utilised to add an additional key step as an anti-counterfeiting agent, or as a low-temperature switch.

## Results

**Physical characterization.** **LaL1(TTA)$_3$** (Fig. 1e) and another control complex, **LaPhen(TTA)$_3$** ((1,10-phenanthroline)tris[4,4,4-trifluoro-1-(2-thienyl)-1,3-butanedionato] lanthanum(III), Supplementary Fig. 1) were synthesized as described in the Methods Section. The $^1$H and $^{13}$C NMR spectra (Supplementary Fig. 2) were recorded in toluene-$d_8$ and chloroform-$d$ for **LaL1(TTA)$_3$** and **L1**. The lanthanide ion complex **LaL1(TTA)$_3$** exhibits sharp, well-resolved spectra in both solvents, demonstrating stable coordination of the ligands with the La$^{3+}$ ion. The ligand **L1** peaks are observed in the FT-IR spectrum of **LaL1(TTA)$_3$** whereas the

carbonyl stretching mode of **TTA** is shifted to lower energy in **LaL1(TTA)$_3$** showing the involvement in complex formation with La$^{3+}$ (Supplementary Fig. 3e).

**Photophysical properties.** The emission and absorption electronic spectra of **LaL1(TTA)$_3$** do not exhibit significant change with concentration at room temperature (Supplementary Fig. 3). However, the visual characteristics of the emission from **LaL1(TTA)$_3$** are remarkable. Upon irradiation at 77 K, 10 μM **LaL1(TTA)$_3$** in toluene shows a bright green afterglow which was observable up to 30 s by the naked eye under 355 nm laser excitation (Fig. 1g; Supplementary Movie 1). The bright green afterglow is identified with the emission peak at 505 nm (Fig. 2a, b). Under the same conditions, another La$^{3+}$ complex, **LaPhen(TTA)$_3$**, exhibits a shorter phosphorescence decay and the emission is only observable up to 2 − 3 s by the naked eye while that of **L1** is observed above 5 s (Fig. 1g; Supplementary Movie 2). Long-lived phosphorescence can also be seen under white light excitation for **LaL1(TTA)$_3$** at 100 μM concentration (Supplementary Movie 3), which further widens the field of application for this compound.

The 77 K emission spectrum of **LaL1(TTA)$_3$** at 10 μM concentration in toluene (Fig. 2a, b) is more clearly resolved than the room temperature spectra (Supplementary Fig. 3c). It shows a longer wavelength singlet vibronic structure and in addition triplet emission is observed with the peak maximum at 505 nm. The relative intensity of these bands depends upon the method of excitation and detection (Supplementary Fig. 4c). However, Fig. 2a demonstrates a dramatic change in the emission spectrum of **LaL1(TTA)$_3$** in toluene at 77 K when the concentration is raised above 10 μM. A reversible interconversion occurs (from form A) while cooling to another form (form B) which has the singlet emission transition with maximum intensity at 469 nm. The triplet T$_1$ → S$_0$ transition of form B is observed at 540 nm. The reversible interconversion occurs whilst cooling and both forms may be present at 77 K depending upon the concentration in toluene (Supplementary Fig. 5a). The difference in the excitation spectra of the respective singlet peaks of Forms A and B (Supplementary Figs. 4g-i) confirms that the change is not due to excimer formation but occurs in the electronic ground state. A similar emission spectrum to that of **LaL1(TTA)$_3$** at 100 μM concentration in toluene is achieved at a lower concentration by changing to the more polar solvent chloroform (Fig. 2b; Supplementary Fig. 4b). Conformational equilibria depending upon temperature and solvent polarity are well-documented in the literature[21,22] and in the present case (Supplementary Fig. 4f) it is noted, that similar concentration-dependent emission spectra are found for the ligand **L1** alone (Supplementary Fig. 6c).

The relatively sharp emission peak of **LaL1(TTA)$_3$** in toluene at room temperature is red-shifted to give a broad singlet peak at 540 nm in the solid-state (Fig. 2c). Based on such considerations, and similar to other reported red-shifted emission spectra in the solid state[23,24], form B can be assigned to the π−π aggregated form while form A is the isolated **LaL1(TTA)$_3$** complex. Turning to time-dependent density functional theory (TD-DFT), the first singlet state transition, S$_1$ – S$_0$, of the π−stacked conformation of **L1** is calculated to be at lower energy compared to the isolated molecule **L1** (Supplementary Fig. 7c), both of which involve a transition to the triazine fragment of **L1**. This phenomenon is more marked in the case of **LaL1(TTA)$_3$** where the −TTA fragment is participating in the energy transition (Fig. 2d). The involvement of TTA is further demonstrated by the −TTA facilitated white light generation with the analogous Eu$^{3+}$ solid compound in the dominant solid-state form B[25].

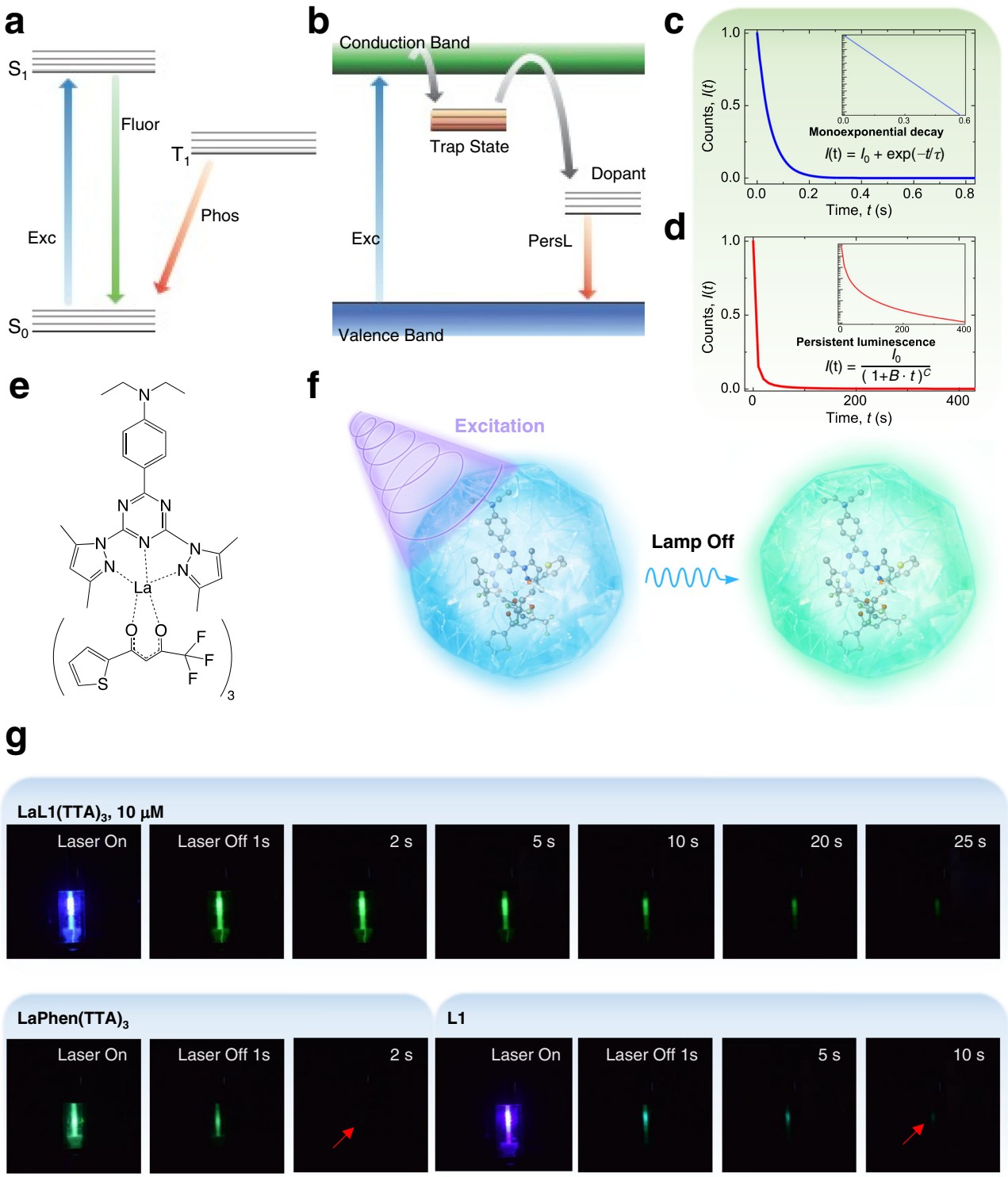

**Fig. 1 Differences between phosphorescence and persistent luminescence and performance of LaL1(TTA)₃.** The schematic Jablonski diagram of (**a**) fluorescence (Fluor) and phosphorescence (Phos) of an organo-lanthanide complex upon excitation (Exc) with the absence of energy transfer to the trivalent lanthanide ion. Phosphorescence usually only occurs under cryogenic condition due to the high nonradiative rate $T_1 \rightarrow S_0$ at room temperature. (**b**) Persistent luminescence (PersL) may occur when trapped electrons (or holes) transfer to the lanthanide dopant ions through thermal release via the conduction band or tunneling. (**c**) The ideal exponential luminescence decay of phosphorescence with lifetime 0.05 s. (**d**) Ideal hyperbolic decay of persistent luminescence with rate constant $B = 0.25\,s^{-1}$. The insets of (**c**) and (**d**) display ordinates on logarithmic scales. (**e**) The chemical structure of **LaL1(TTA)₃**. (**f**) The schematic illustration with a photograph of the long-lived afterglow phosphorescence of **LaL1(TTA)₃** after 355 nm excitation. (**g**) The long-lasting afterglow of **LaL1(TTA)₃** at 10 μM concentration in toluene is observable for up to 30 s after Nd³⁺:YAG laser irradiation at 77 K. The afterglow of **LaPhen(TTA)₃** (diminished after 2 to 3 s) and of **L1** (just above 5 s) are shown for comparison. (**L1** = 2-(N,N-diethylanilin-4-yl)−4,6-bis(3,5-dimethylpyrazol-1-yl)-1,3,5-triazine (dbpt); **TTA** = thenoyltrifluoroacetonate; **Phen** = 1, 10-phenanthroline).

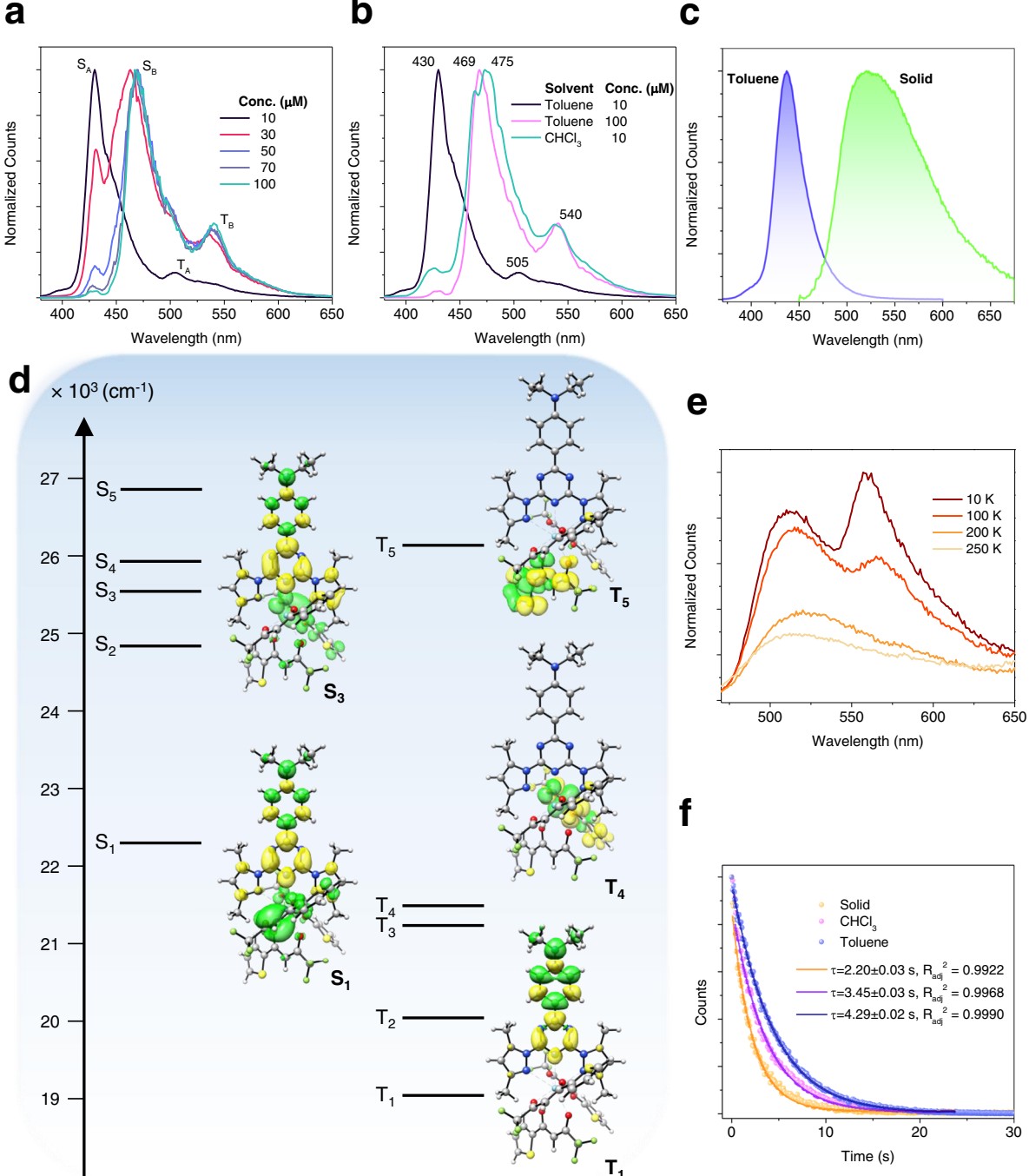

**Fig. 2 Emission spectra using 355 nm excitation.** (**a**) The emission spectrum of **LaL1(TTA)₃** in toluene at different concentrations at 77 K. (**b**) The comparison of the 77 K emission spectrum of **LaL1(TTA)₃** at different concentrations in toluene and in CHCl₃. The emission spectrum of **LaL1(TTA)₃** in toluene at high concentration resembles that of 10 μM concentration in CHCl₃. (**c**) The comparison between the emission spectrum in toluene (10 μM) and that in the solid-state at room temperature. (**d**) The optimized structure of **LaL1(TTA)₃** and the energy diagram showing the computed TD-DFT energy levels, with corresponding orbital transitions. The ordinate is energy. The green colour of the orbitals represents the decrease in occupation during the electronic transition whilst that of yellow represents an increase. (**e**) The variation of the solid-state emission spectrum with temperature. The triplet peak at 560 nm is barely observed above 100 K. (**f**) Triplet state decays of **LaL1(TTA)₃** in CHCl₃ (10 μM), toluene (10 μM) at 77 K and in the solid-state at 10 K. The fitted decay lifetimes are indicated with the adjusted coefficients of determination.

Upon cooling solid **LaL1(TTA)₃** from 250 K to 10 K, a new band at 560 nm is observed (Fig. 2e; Supplementary Fig. 5b). The peak disappears when the sample is heated up above 100 K, illustrating that this band corresponds to the triplet state of form B, which quenches at higher temperatures.

The phosphorescence lifetimes of **L1** and **LaL1(TTA)₃** at 77 K were determined from monoexponential fits to the decay profiles and were 0.18 s for **L1** and between 3.5 s and 4.3 s for **LaL1(TTA)₃**, depending upon the solvent (Fig. 2f). The lifetime

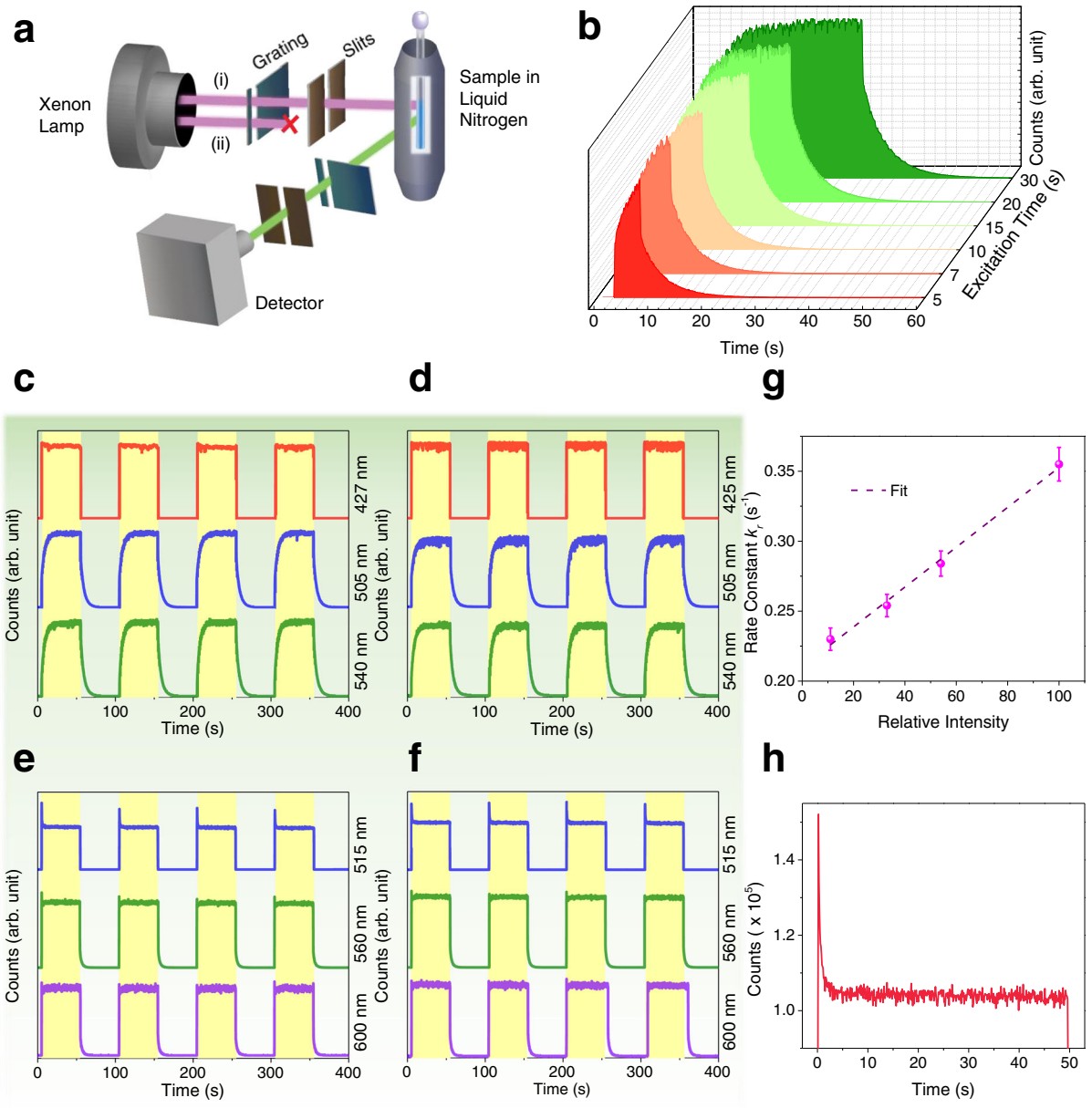

**Fig. 3 Kinetics Profile of triplet and singlet transitions of LaL1(TTA)₃ at 77 K (or 10 K where stated).** (**a**) Schematic setup of Horiba kinetics module. Two different modes were used with (i) the luminescence was monitored while the irradiation remained and (ii) the slit for irradiation was closed and only the luminescence after excitation was monitored. (**b**) The time-resolved intensity profile evolution of triplet emission (505 nm) of **LaL1(TTA)₃** with different excitation duration using 355 nm. Kinetics charging and decay profiles of **LaL1(TTA)₃** at 10 μM concentration in toluene using (**c**) 355 nm and (**d**) 395 nm excitation. (**e**),(**f**) Profiles for solid-state **LaL1(TTA)₃** at 10 K using (**e**) 355 nm and (**f**) 395 nm excitation. In each case for (**c**)-(**f**), the monitored wavelengths are indicated on the right-hand side. (**g**) Dependence of measured rise rate constant ($k_r$) upon charging intensity for **LaL1(TTA)₃** at 10 μM concentration in toluene. The dashed line is fitted with the adjusted coefficient of determination $R_{adj}^2 = 0.9928$. Error bars represent the standard deviations of the averaged fitted rise rate constant. The decay constant ($k_d$) is effectively constant (Supplementary Fig. 4k). (**h**) Expanded view of the initial part of the third cycle of (**e**) monitoring 515 nm emission.

in the solid-state was shorter, being 2.2 s at 10 K, and decreased upon increasing the temperature (Fig. 2f; Supplementary Fig. 5c). For comparison, the triplet state lifetime of **LaPhen(TTA)₃** was measured as 0.18 s, which is shorter than the phosphorescence decay lifetime of **Phen** itself (1.6 s[26]). Hence the longer phosphorescence lifetime of **LaL1(TTA)₃** compared to that of **LaPhen(TTA)₃** does not solely arise from metal coordination but involves other photophysical processes.

**Time-resolved emission intensity properties of LaL1(TTA)₃.** Further characterization of **LaL1(TTA)₃** was performed using the kinetics module as illustrated in Fig. 3a. The emission counts at 77 K were monitored at selected wavelengths using different excitation wavelengths, by switching the xenon source sequentially on and off at fixed time intervals. The time-resolved intensity profile at 505 nm of **LaL1(TTA)₃** at 10 μM concentration in toluene at 77 K revealed a distinct charging feature for this

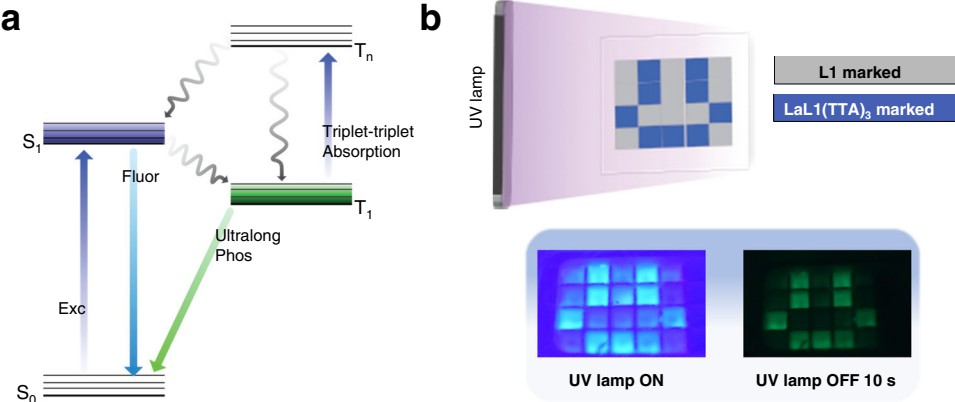

**Fig. 4 Photophysical mechanism and application of LaL1(TTA)$_3$.** (**a**) The Jablonski diagram illustrates the spectroscopic processes for LaL1(TTA)$_3$. The charging can be explained by the spin-allowed $T_1 \rightarrow T_n$ absorption process followed by the $T_n$ to $S_1$ and $T_1$ processes. (**b**) The application of the long-lasting phosphorescence of **LaL1(TTA)$_3$** in encryption under cryogenic conditions. The top diagram depicts the experimental arrangement.

sample (Fig. 3b), which leads to saturation at longer charging times.

Using 355 nm and 395 nm as excitation wavelength, the nearly-flat kinetics profile of 427 nm singlet charging and emission (Fig. 3c) arises because the lifetime of this $S_1 \rightarrow S_0$ transition is fast (about 1.5 ns). The 505 nm and 540 nm emission bands correspond to triplet emission (Fig. 2b), with the 540 nm band being part of the vibronic sideband of the same transition. Considering that form A dominates at 10 μM concentration, the profiles of these two emission bands (Fig. 3c, d) are the same under both irradiation wavelengths and exhibit charging with a monoexponential rise lifetime of 4.3 s. As discussed beforehand, form B (rather than form A) of **LaL1(TTA)$_3$** mainly exists at 100 μM concentration in toluene. When monitoring the phosphorescence decay at different concentrations of **LaL1(TTA)$_3$** in toluene, the rise lifetime ($\tau_r$) at 505 nm (corresponding to $\tau_{A,r}$) did not change much with concentration, while the lifetime monitored at 540 nm (corresponding to $\tau_{A,r}$ and/or $\tau_{B,r}$) decreases with increasing concentration due to the change in relative amounts of A and B (Supplementary Fig. 4j). The difference in lifetime confirms that A and B correspond to different species, regardless that both demonstrate long-lived phosphorescence. The corresponding decay kinetics curves for both the 10 μM (Fig. 3c, d) and 100 μM concentrations (Supplementary Fig. 9a, b), are monoexponential (Supplementary Fig. 10a, b) with decay lifetimes ($\tau_d$) slightly longer than the rise (charging) lifetimes ($\tau_r$). The charging profile can be attributed to triplet-triplet absorption involving $T_1$, where the decay lifetime is always longer than the charging lifetime as derived in Methods, Eq. (5). The decay lifetime is independent of the charging process and is constant within experimental error (Supplementary Fig. 4k). Moreover, the charging (rise) lifetime decreases with increasing excitation intensity (Supplementary Fig. 4k). This is as expected for the triplet-triplet absorption of $T_1$. The detailed solution of the rate equations laid out in the Methods shows that the inclusion of triplet-triplet absorption leads to $k_r > k_d$ for the rise ($k_r$) and decay ($k_d$) rate constants and that $k_r$ should exhibit a linear relation with charging intensity (Supplementary Fig. 10h). This is borne out in Fig. 3g. The rise profiles of $S_1$ and $T_1$ calculated from the Rosenbrock numerical method (Supplementary Fig. 10d, e) are in reasonable agreement with experimental data. We have not attempted to fit the experimental data because only the values for parameters $b$ and $d$ are available from our experiments.

The observations when employing solid **LaL1(TTA)$_3$**, where π-stacking interaction is much stronger, differ somewhat from those in toluene. However, when monitoring at 427 nm (Fig. 3c) or

515 nm (Fig. 3e), in both cases a sharp cut-off is observed when illumination is turned off, as expected for the decay of a singlet transition. The singlet state kinetics profile for the solid demonstrates peculiar properties which are reported herein (Fig. 3e, f, h). Upon exciting with 355 nm or 395 nm radiation and monitoring the singlet emission band at 515 nm, an initial sharp increase in intensity is followed by decay to a plateau - although the sample is continuously excited. The initial spike is scarcely observed for 10 μM **LaL1(TTA)$_3$** in toluene (Fig. 3c and Supplementary Fig. 10e). The spike is attributed to the fast singlet excitation process followed by a slow intensity decrease due to intersystem crossing to $T_1$. The decay to a plateau (with the rate constant value equal to the $T_1$ rise rate constant) indicates a successive reduction in the population of the singlet state until all processes are in equilibrium. Since the volume available per formula unit is reduced by a factor of ~1 million on going from 10 μM **LaL1(TTA)$_3$** in toluene to solid-state **LaL1(TTA)$_3$**, the heavy atom effect of $La^{3+}$ in increasing the intersystem crossing rate via spin-orbit coupling is more pronounced in the solid-state. This is demonstrated in Supplementary Fig. 10g for the calculated $S_1$ charging profile, where the only change from the parameters employed in Supplementary Fig. 10e is the increase of intersystem crossing rate by the factor of 100. The approach to the equilibrium plateau in the $S_1$ charging profile is much faster in the solid-state, followed by a greater intensity decrease. The schematic photophysical mechanism for **LaL1(TTA)$_3$** at 10 μM concentration in toluene is illustrated in Fig. 4a.

**Real life application of LaL1(TTA)$_3$.** We took further steps to demonstrate the potential application of this compound, particularly for encryption purposes (Fig. 4b). Different logos or passwords in squares can be filled with **LaL1(TTA)$_3$** in an appropriate vessel while the remaining squares are filled with a blue-emitting compound (**L1** is used herein). The vessel shows blue emission from all squares at room temperature (Supplementary Movie 4). However, the desirable logo can be captured after excitation with a UV lamp under the cryogenic state (Fig. 4b; Supplementary Movie 5). It should also be noted that only a low-power UV lamp (4 W) is needed for this application and that miniaturization is simple.

In order to extend the triplet rise time duration, for a fixed triplet decay lifetime and irradiation power, we have found that several processes are most important, as described by our rate equation calculations. First, an increase in $S_1 \rightarrow T_1$ intersystem crossing rate shortens the $T_1$ charging lifetime (Supplementary Fig. 10i). Second, the $T_1$ charging rise lifetime increases when the

internal conversion rate from $T_n \rightarrow T_1$ increases. Third, the $T_1$ rise lifetime notably shortens when the $T_n \rightarrow S_1$ intersystem crossing rate increases. It is interesting to find appropriate chemical manipulations that would enable these rates to be tuned.

In conclusion, we have synthesized and reported the lanthanide compound with a long phosphorescence lifetime which gives emission observable up to 30 s by the naked eyes at 77 K. This observation can be facilitated in various fields, including counterfeiting and encryption agents, as demonstrated herein. The peculiar charging and decay kinetics properties demonstrated by this compound are elucidated by the presence of triplet-triplet absorption. Significantly, this study provides not only a potential agent for long-lasting phosphorescence under the cryogenic state, but the kinetic analysis also demonstrates the fundamental criteria to be optimized for long-lasting phosphorescence, be it at room temperature or in the cryogenic state.

## Methods

**Materials**. All reagents and all anhydrous solvents were purchased from Sigma Aldrich. Unless otherwise noted, all chemicals and solvents (Duksan) were used as received without further purification.

La(TTA)$_3 \cdot$2H$_2$O and **L1** were prepared by the literature methods without modification[27,28].

**Synthesis of L1**. A mixture of N,N-diethylaniline (16 g, 0.108 mol) and cyanuric chloride (9.88 g, 0.054 mol) was heated at 75 °C for 24 h under nitrogen. After complete reaction, the mixture was extracted with 100 mL dichloromethane (30 mL × 3). The combined organic phase was dried over anhydrous MgSO$_4$, then filtrated through celite and washed with dichloromethane (10 mL × 3). The resulting mixture was evaporated under vacuum and the solid residue was washed with 50 mL × 3 of n-hexane. Finally, the intermediate 4-(4,6-dichloro-1,3,5-triazin-2-yl)-N,N-diethylaniline was afforded from twice recrystallization in acetone as a yellow solid (4.01 g, 25%). Potassium (271 mg, 6.96 mmol) was added to a stirred solution of 3,5-dimethylpyrazole (970 mg, 95 mmol) in dry THF (60 mL) under N$_2$. The resulting mixture refluxed until a colourless solution was formed. Subsequently, 4-(4,6-dichloro-1,3,5-triazin-2-yl)-N,N-diethylaniline **2** (1 g, 3.365 mmol) was added to aforementioned solution at about 10 °C. The reaction mixture was further stirred at room temperature for 0.5 h and then refluxed at 70 °C for 6 h. After complete reaction, the mixture was kept at 4 °C for 12 h (overnight) to ensure a complete precipitation. The precipitate was finally filtered, washed repeatedly with THF (10 mL × 3) and n-hexane (10 mL × 3). The solid was dried under vacuum to afford **L1** as a yellow solid (0.96 g, 63%).

**Synthesis of La(TTA)$_3\cdot$2H$_2$O**. La(TTA)$_3 \cdot$2H$_2$O was synthesized by the literature method[29] with some modifications. Briefly, HTTA (0.67 g, 3 mmol) was dissolved in 15 mL ethanol in a 150 mL flask, with stirring at room temperature. Then, the pH of the solution was adjusted to 6–7 by the addition of NaOH solution (1.0 M). After that, 1 mmol of LaCl$_3 \cdot$7H$_2$O (0.37 g, 1 mmol) solution in 5.0 mL of deionized water was then added into the above mixture at 60 ºC. Then deionized water (100 mL) was added into the above mixture with vigorous stirring for 2 h at 60 ºC to ensure complete precipitation. After cooling to room temperature, the precipitate was filtered, washed repeatedly with water, and dried overnight under vacuum at room temperature to afford La(TTA)$_3 \cdot$2H$_2$O as a light yellow solid (0.71 g, 85%).

**Synthesis of LaL1(TTA)$_3$**. The solution of the **L1** (41.6 mg, 0.1 mmol) and La(TTA)$_3 \cdot$2H$_2$O (86.0 mg, 0.1 mmol) in THF (20 mL) was stirred at room temperature overnight. The solvent was then evaporated and the residue was redissolved in a minimum amount of diethyl ether and precipitated by n-hexane. The bright yellow solid was obtained by twice repeated precipitation.

Yield: 71 %. $^1$H NMR (400 MHz, Chloroform-d) δ 8.24 (d, J = 8.8 Hz, 2H), 7.48 (d, J = 3.7 Hz, 3H), 7.37 (d, J = 5.0 Hz, 3H), 7.00 – 6.93 (m, 3H), 6.70 (d, J = 8.9 Hz, 2H), 6.13 (s, 3H), 6.11 (s, 2H), 3.43 (q, J = 7.1 Hz, 4H), 2.88 (s, 6H), 2.49 (s, 6H), 1.22 (t, J = 7.0 Hz, 6H); $^{13}$C NMR (101 MHz, Chloroform-d) δ 180.17, 172.60, 169.88 (q, J = 31.9 Hz), 162.41, 155.90, 152.23, 146.61, 145.05, 132.01, 131.03, 128.76, 127.58, 120.20, 119.12 (q, J = 287.0 Hz), 113.31, 111.00, 91.56, 44.82, 16.91, 13.68, 12.62; $^1$H NMR (400 MHz, Toluene-d$_8$) δ 8.10 (d, J = 8.8 Hz, 2H), 7.17 (dd, J = 3.8, 1.1 Hz, 3H), 6.80 (dd, J = 5.0, 1.1 Hz, 3H), 6.49 (dd, J = 5.0, 3.7 Hz, 3H), 6.45 (s, 3H), 6.40 (d, J = 8.8 Hz, 2H), 5.63 (s, 2H), 2.81 (d, J = 6.9 Hz, 4H), 2.79 (s, 6H), 2.54 (s, 6H), 0.80 (t, J = 7.0 Hz, 6H); $^{13}$C NMR (101 MHz, Toluene-d$_8$) δ 180.60, 172.29, 170.13 (q, J = 31.3 Hz), 162.15, 155.58, 151.87, 146.57, 144.76, 131.92, 130.80, 127.40, 120.63 (d, J = 287.1 Hz), 120.49, 113.04, 110.68, 91.74, 44.13, 16.45, 13.75, 12.13. Elemental analysis (%, calcd., found): C(46.31, 46.64), H(3.31, 3.28), N (9.19, 9.27), S(7.89, 7.31).

**Synthesis of LaPhen(TTA)$_3$**. The LaPhen(TTA)$_3$ samples were synthesized by the literature method[30]. Typically, TTA (3 mmol) and Phen (1 mmol) were dissolved in 15 mL ethanol in a flask with stirring, at room temperature. Then, the pH of the solution was adjusted to 7.0 by the addition of NaOH solution (1.0 M). After that, 1 mmol of LaCl$_3$ solution in 5.0 mL of deionized water was then added into the above mixture at 60 °C with vigorous stirring for 1.0 h to ensure complete precipitation. The precipitate was finally filtered, washed repeatedly with ethanol and water, and dried overnight under vacuum.

Yield: 65%; HRMS (ESI) calculated for C$_{24}$H$_{12}$F$_9$LaNaO$_6$S$_3$$^+$ [La(TTA)$_3$ + Na]$^+$ 824.8608, found 824.8608. HRMS (ESI) calculated for C$_{24}$H$_{14}$F$_9$LaO$_8$S$_3$$^-$ [La(TTA)$_3$ + COO]$^-$ 847.8698, found 847.8679[31]. Elemental analysis (%, calcd., found): C (44.00, 44.13), H (2.05, 2.77), N (2.85, 2.50).

**Sample characterization**. Both $^1$H and $^{13}$C NMR spectra were collected on a Bruker Ultrashield 400 Plus 400 MHz NMR spectrometer. High-resolution mass spectra (HRMS) were obtained on an Agilent 6540 UHD AccurateMass Q-TOF LC/MS. Elemental analyses were performed by an Elementar microcube elemental analyzer. FT-IR ATR spectra were recorded by a Perkin Elmer® FT-IR Spectrum Two instrument.

**Photophysical characterization**. The ultraviolet-visible absorption spectra were measured in solution in the range 200–800 nm by a Perkin Elmer® LAMBDA 1050+ UV/VIS/NIR double beam spectrophotometer. The emission, excitation spectra, and luminescence decay curves were recorded using a Horiba Fluorolog®-3 instrument with a 450 W xenon lamp. The signal was detected by a Hamamatsu R928 photomultiplier and corrected with excitation and emission correction factors to eliminate response characteristics. For laser irradiation, a Nd$^{3+}$:YAG pulsed laser was used as the excitation source together with a third harmonic generator (10 mJ) and an optical parameter oscillator (OPO, Spectra-Physics versaScan, and UVScan; pulse duration 8 ns, repetition frequency 10 Hz). A 10 mm pathlength cuvette was used for all room temperature measurements. A custom-made liquid nitrogen cryostat with an NMR tube sample holder was employed for 77 K studies. The Horiba kinetics module as shown in Fig. 4 was employed for time-resolved intensity studies. The sample was housed in a CS202-DMX-1AL closed-cycle cryostat from Advanced Research Systems for low temperature (nominal 10 K temperature) studies. A qX3/Horiba4 variable temperature cell was employed to investigate the temperature dependence of emission between 280 to 323 K.

**DFT calculations**. DFT calculations were performed using the ORCA 4.0 suite of programs[32]. Molecules were built using Avogadro software[33]. The integration grid was set to Lebedev 590 points with the final grid of Lebedev 770 points. The nature of each stationary point was characterized using frequency calculations. Geometry optimizations and free-energy calculations were performed at the PBE0-D3/def2-TZVP level of theory[34], with the Stuttgart in-core effective core potentials and basis sets for the lanthanide ions[35,36]. Solvation by toluene (ε = 8.93) was accounted for using the conductor-like polarizable continuum model (CPCM)[37]. TD-DFT calculations were performed with the same level of theory using the Pople solver. The simulated absorption spectra employed the bandwidth of 3000 cm$^{-1}$. UCSF Chimera (version 1.14)[38] was used for the visualization of geometries and orbitals.

**Kinetics data model**. We consider the four-level system comprising S$_0$, S$_1$, T$_1$, and T$_n$. The rate constants (s$^{-1}$) are as defined in Table 1.

**Table 1 Rate constants and processes in the four level system.**

| Rate constant | Process | Meaning |
|---|---|---|
| a | S$_0$ + hν → S$_1$ | Excitation |
| b | S$_1$ → S$_0$ | Radiative and nonradiative relaxation |
| c | S$_1$ → T$_1$ | Intersystem crossing |
| d | T$_1$ → S$_0$ | Radiative and nonradiative relaxation |
| f | T$_1$ + hν → T$_n$ | Triplet-triplet excitation |
| g | T$_n$ → T$_1$ | Internal conversion |
| h | T$_n$ → S$_1$ | Intersystem crossing |
| k | T$_n$ → S$_0$ | Radiative and nonradiative relaxation |

The excitation rate (s$^{-1}$) $a = \sigma I_{exc}/h\nu$ depends on the absorption cross-section σ (m$^2$) of S$_0$ → S$_1$, the excitation intensity, $I_{exc}$ (W m$^{-2}$) and on the excitation photon energy, hν (J). Since both the S$_0$ → S$_1$ and T$_1$ → T$_n$ transitions are spin-allowed we use similar values for a and f.

The rate equations are:

$$\frac{d[S_0]}{dt} = -a[S_0] + b[S_1] + d[T_1] + k[T_n] \tag{1}$$

$$\frac{d[S_1]}{dt} = a[S_0] - (b+c)[S_1] + h[T_n] \tag{2}$$

$$\frac{d[T_1]}{dt} = c[S_1] - (d+f)[T_1] + g[T_n] \tag{3}$$

$$\frac{d[T_n]}{dt} = f[T_1] - (g+h+k)[T_n] \tag{4}$$

A simple solution, considering the estimated values of rate constants, when applying the steady-state approximation to $S_1$ and $T_n$ (since decay of $T_1$ is so slow) gives:

$$[T_1] \sim A[1 - \exp(-(k_r t)] \tag{5}$$

where $A$ is a constant and the rise exponent $k_r > c = k_d$, where $k_d$ is the decay rate constant of $T_1$ (i.e., $\tau_r < \tau_d$). The fit to the experimental data using this equation is shown in Supplementary Fig. 10c. Note that the transfer to the solvent toluene lowest triplet state (345 nm) is not possible.

The numerical solution to Eqs. (1)–(4) using Maple 2021$^{TM}$ by the Rosenbrock stiff 3-4th order numerical method (or the Modified Extended Backward Differentiation Equation Implicit Method) with initial conditions that $[S_0](0) = 1$, $[S_1](0) = 0$, $[T_1](0) = 0$ and $[T_n](0) = 0$ is straightforward and representative results are given in the Supplementary Fig. 10d–i with the parameter set shown.

## Data availability

All data generated or analysed during this study, which support the plots within this paper and other findings of this study, are included in this published article and are available upon reasonable request to the authors.

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

## Acknowledgements

Financial assistance from the Hong Kong Research Grants Council Grant No. 12300021 and the Centre for Medical Engineering of Molecular and Biological Probes (AoE/M-401/20) is gratefully acknowledged (K.-L.W.). We thank the Max-Planck-Institut für Chemische Energiekonversion for making available the Orca Program, and Mr. Karel Au is thanked for constructing a computer to run ORCA in the Linux platform. This research was conducted using also the resources of the High-Performance Cluster Computing Centre, Hong Kong Baptist University, which receives funding from the Research Grants Council, University Grant Committee of the HKSAR, and Hong Kong Baptist University. We thank Zhaolun Song of Maplesoft for useful suggestions and Sci-Hub in removing barriers in the way to science.

## Author contributions

K.-L.W. conceived and oversaw the project. Y.W., L.W., and Y.Z. performed syntheses and NMR studies. W.T. performed the experimental work. P.A.T. and W.T suggested experiments, carried out calculations, and jointly wrote the paper.

## Competing interests

The authors declare no competing interests.
