## [Peer Review File · Nature Communications]

Charging and Ultralong Phosphorescence of Lanthanide Facilitated Organic ComplexREVIEWER COMMENTS

Reviewer #1 (Remarks to the Author):

Thor et. al. report a lanthanide coordination complex LaL1(TTA)_3 exhibiting remarkable phosphorescence observable up to 30s with the unaided eye and interesting charging behavior under 77 K. The authors attributed this novel optical phenomenon to triplet-triplet absorption. Moreover, intermolecular interactions make a great contribution to the photophysical properties of LaL1(TTA)_3 . Specifically, a concentration-dependent emission is found along with the interconversion between isolated molecules and π stacking layers, and the fluorescence intensity of the solid sample reveals an initial spike due to the delocalization of the excitation from monomer to an extended species. There is no doubt that the paper presents a valuable phenomenon for both fundamental study and optical devices. I support the publication of this excellent work. However, there are still some important issues that must be addressed.

1. In line 91, the authors claim that the stable coordination of La^{3+} and L1 is confirmed by NMR data. However, according to the mass spectra of LaPhen(TTA)_3 , the coordination bond between La^{3+} and Phen seems not very strong. So, I am confused about how to identify whether the NMR peaks of L1 are attributed to coordination complexes or free molecules dissociated from LaL1(TTA)_3 ? More details should be explained.

2. The cryogenic sample is not well-characterized. For a high-concentration solution, cooling the sample may lead to phase separation or precipitation. Thus, the phase stability needs to be discussed, the authors are supposed to supply some other evidence, such as the photo image and DSC data. Also, the reproducibility of those optical properties should be checked for several cooling-heating cycles.

3. The authors believe that charging and decay properties are attributed to the triplet-triplet absorption, it might be reasonable, but other in-situ experimental evidence, such as transient absorption spectrum, might be useful to provide more evidence.

4. The interconversion between form A and form B is an interesting phenomenon. An ambiguous description of form B seems to be a moderate π - π aggregated state between monomer and solid states. But the authors should give more specific discussion about the conformation of form B, such as the stacking mode or π - π interaction distance. I am also confused about the structural difference between form B and solid states, and whether form B will be transformed to solid states at high concentration?

5. The reported solid sample is precipitated from diethyl ether/n-hexane, whereas for aggregated states, solvents play an important role in the crystallization process. More information about the novel optical phenomenon may be obtained if the authors could recrystallize the sample from toluene or chloroform.

Minor issues:

a) There are mistakes between the Figures and their description in the manuscript, such as Fig. 2b, Fig. 2f, and Fig. 3b.

b) Full names of LaL1(TTA)_3 , LaPhen(TTA)_3 , and L1 should be added when they first appeared in the text. The structure of LaPhen(TTA)_3 should also be presented.

Reviewer #2 (Remarks to the Author):

The manuscript (Ref: NCOMMS-21-25426-T) of the paper entitled "First Reported Charging and Ultralong Phosphorescence of Lanthanide Facilitated Organic Complex ", in my opinion should be considered for publication in the journal: Nature Communications after its minor revision. The article shows very interesting and novel experimental-theoretical studies concerning development of the first ultra-long phosphorescent lanthanide-based complex, operating in the time scale of seconds. The work is very well executed and the performed research are on the top level. However, the following points should be addressed before publication of the article:

- Have the luminescence spectra been corrected for the apparatus response? This information should be provided in the characterization section
- Figure 1c and d: the Authors may consider presentation of the decay profiles in the exponential representation (Y-axis; intensity scale)
- Figure 3b: maybe a bigger offset/different perspective could be applied to better show the rise (charging) profiles. The most important seem to be the first five profiles, as in the last two ones (20 and 30 s excitation time) we clearly observe the saturation effect, which in fact is not commented by the Authors
- Figure 3c-f: : increase the font size in the x-axes (enlarge the numbers)
- Figure 7a and b in Extended data: increase the font size in the x-axes (enlarge the numbers)
- Characterization section: the type (e.g. OPO) and detailed technical parameters (pumping source, pulse duration, energy, etc.) of the pulsed laser used should be provided in the characterization section

Reviewer #3 (Remarks to the Author):

Wong et al reported long phosphorescence of lanthanide complexes, and the 'charging' feature of the system. Long luminescence from a Ln complexes, or phosphorescent organic chromophore is not new. Long phosphorescence lifetime from an organic chromophore, especially at low temperature, up to seconds, is well known in photochemistry.

Some sentences are confusing, such as 'Intersystem crossing processes from S1 to T1 are spin-forbidden but the rates are relaxed upon coordination to a heavy metal, such as for lanthanide complexes', what does it mean for a 'rate' to 'relax'?

Figure 4a was used to explain the 'charging' feature of the molecules. From a point of view of photochemistry, this is impossible. The T_n state should be with very short lifetime, due to the fast internal conversion (IC) to T1 state. The mechanism is not convincing.

Response to Reviewers' Comments

Comments from the Reviewer 1:

Thor et. al. report a lanthanide coordination complex LaL1(TTA)_3 exhibiting remarkable phosphorescence observable up to 30s with the unaided eye and interesting charging behavior under 77 K. The authors attributed this novel optical phenomenon to triplet-triplet absorption. Moreover, intermolecular interactions make a great contribution to the photophysical properties of LaL1(TTA)_3 . Specifically, a concentration-dependent emission is found along with the interconversion between isolated molecules and π stacking layers, and the fluorescence intensity of the solid sample reveals an initial spike due to the delocalization of the excitation from monomer to an extended species. There is no doubt that the paper presents a valuable phenomenon for both fundamental study and optical devices. I support the publication of this excellent work. However, there are still some important issues that must be addressed.

Comment 1: In line 91, the authors claim that the stable coordination of La^{3+} and L1 is confirmed by NMR data. However, according to the mass spectra of LaPhen(TTA)_3 , the coordination bond between La^{3+} and Phen seems not very strong. So, I am confused about how to identify whether the NMR peaks of L1 are attributed to coordination complexes or free molecules dissociated from LaL1(TTA)_3 ? More details should be explained.

Response: Thank you for your suggestions. Concerning LaPhen(TTA)_3 , the HRMS (ESI) data given in the Synthesis section were intended for characterization only and do not indicate bond strength between La^{3+} and Phen. We did not record the spectrum to higher mass since the bonding of Phen nitrogen to the lanthanide ion has already been demonstrated by HRMS (ESI) data.¹ To confirm the coordination of the ligands to the metal complexes, we can observe the shifting of the peaks when comparing the NMR spectra of L1 and the LaL1(TTA)_3 complex (Supplementary Fig. 2).

The Fourier transform infrared (FT-IR) spectrum of LaL1(TTA)_3 (Fig. 1 below) shows the peaks of L1. The carbonyl stretching vibration in TTA shifts to lower wavenumber showing the involvement of the carbonyl oxygen in complex formation to La^{3+} .

1. H. Gallardo *et al.* Synthesis, structure and OLED application of a new europium(III) complex: {tris-(thenoyltrifluoroacetate)[1,2,5]selenadiazolo[3,4-f][1,10]phenanthroline}europium(III). *Inorg. Chim. Acta* **473**, 75–82 (2018).

Figure 1. The FT-IR spectra of **TTA**, **L1** and **LaL1(TTA)₃**.

Comment 2: The cryogenic sample is not well-characterized. For a high-concentration solution, cooling the sample may lead to phase separation or precipitation. Thus, the phase stability needs to be discussed, the authors are supposed to supply some other evidence, such as the photo image and DSC data. Also, the reproducibility of those optical properties should be checked for several cooling-heating cycles.

Response: Thank you for your suggestion. The DSC instrument does not permit cryogenic measurements. Photos of **LaL1(TTA)₃** in 10 μM and 100 μM toluene solution were taken and have been attached.

Figure 2. The photo of **LaL1(TTA)₃** at (a) 10 μM and (b) 100 μM in toluene after cooling to 77 K. **LaL1(TTA)₃** at both concentrations shows a clear single phase without any phase separation or precipitation.

From Fig. 2, we can observe that **LaL1(TTA)₃** at both concentrations shows a similar freezing phase. Besides, we believe that the phase stability of our sample is good since the repeated cooling-heating cycles for **LaL1(TTA)₃** using different concentrations (Fig. 3) demonstrate the same optical properties.

Figure 3. The time-resolved emission intensity of (a) 10 μM ($\lambda_{\text{em}} = 505 \text{ nm}$) and (b) 100 μM ($\lambda_{\text{em}} = 540 \text{ nm}$) **LaL1(TTA)₃** in toluene with repeated cooling-heating cycles ($\lambda_{\text{exc}} = 355 \text{ nm}$).

Comment 3: The authors believe that charging and decay properties are attributed to the triplet-triplet absorption, it might be reasonable, but other in-situ experimental evidence, such as transient absorption spectrum, might be useful to provide more evidence.

Response: We agree that by using transient absorption spectrum technique, the triplet-triplet absorption can be observed directly through a positive ΔOD in the spectrum. However, the triplet-triplet absorption region in our study would occur at around 208 nm (47970 cm^{-1}) from the first triplet state at 505 nm (with $\lambda_{\text{exc}} 355 \text{ nm}$). This is out of the range of inspection for usual transient absorption instruments. Furthermore, toluene has absorption in the region shorter than 280 nm and masks other absorption bands. Hence, it is not possible to observe direct experimental evidence of $T_1 \rightarrow T_n$ excited state absorption from the transient absorption spectrum of our system.

Comment 4: The interconversion between form A and form B is an interesting phenomenon. An ambiguous description of form B seems to be a moderate π - π aggregated state between monomer and solid states. But the authors should give more specific discussion about the conformation of form B, such as the stacking mode or π - π interaction distance. I am also confused about the structural difference between form B and solid states, and whether form B will be transformed to solid states at high concentration?

Response: To address the conformation of form B, DFT calculations on the dimer of **L1** were performed. Besides the TD-DFT results discussed in the Supplementary Fig. 7, the stacked conformations of form B have now also been included in Supplementary Fig. 8 as attached below in Fig. 4.

Figure 4. The stacked conformation of **L1** representing the conformation of form B with (a) showing the distance of the π - π ring stacking between **L1** and (b) showing the dihedral angle between the two π rings.

As seen from Fig. 4, the π -stacking happens between the neighbouring triazine-ring and the benzene-ring.

The 77 K emission spectrum of **LaL1(TTA)₃** in toluene at the higher concentration of 1 mM (Fig. 5) is not markedly different from that at 100 μ M. Further observations are precluded due to the poor solubility of **LaL1(TTA)₃** at higher concentration. The reviewer has questioned whether form B could be transformed into “solid states” at higher concentration. At low concentrations, **LaL1(TTA)₃** exists mainly as monomers in frozen solution. Then, at concentrations near 100 μ M, dimers predominate. Further increase in the concentration to the millimolar regime does not significantly change the emission spectrum so that the arrangement of molecules in solution is not markedly changed due to the similar interactions with the solvent environment. The broad solid-state spectrum is the result of inhomogeneous interactions between molecules in the absence of solvent. The volume available per formula unit is decreased by a factor of ~ 1 million on going from 10 μ M solution to the solid state. This enables new interactions between complex molecules to manifest themselves. We do not consider that a further increase of concentration in solution would result in spectral broadening as in the solid state.

Figure 5. The emission spectra of **LaL1(TTA)₃** in toluene with different concentrations at 77 K ($\lambda_{exc} = 355$ nm).

Comment 5: The reported solid sample is precipitated from diethyl ether/n-hexane, whereas for aggregated states, solvents play an important role in the crystallization process. More

information about the novel optical phenomenon may be obtained if the authors could recrystallize the sample from toluene or chloroform.

Response: This is a good point. We recrystallized **LaL1(TTA)₃** from toluene. The photophysical properties are shown in Fig. 6 below and are compared with those of the previously recrystallized sample using the diethyl ether/n-hexane system.

Figure 6. The comparison of the emission spectra of **LaL1(TTA)₃** recrystallized using diethyl ether/n-hexane or toluene: (a) 10 μ M and (b) solid at room temperature. (c) The emission spectra recorded at 10 μ M in toluene at 77 K. (d) The kinetics of **LaL1(TTA)₃** recrystallized from toluene monitored at the triplet state (505 nm) also demonstrate the same long lasting phosphorescence ($\lambda_{exc} = 355$ nm).

In Fig. 6a, **LaL1(TTA)₃** recrystallized from toluene and then dissolved in toluene at 10 μ M concentration shows a similar emission band at room temperature. The emission spectra of solid **LaL1(TTA)₃** at room temperature and 77 K are similar for both recrystallization systems. The sample which was recrystallized using toluene demonstrates the same long lasting phosphorescence, as shown in Fig. 6d.

To further characterize the recrystallized samples, we recorded their FT-IR spectra (Fig. 7).

Figure 7. The FT-IR spectra of **LaL1(TTA)₃** recrystallized from diethyl ether/n-hexane or toluene.

The FT-IR spectra of **LaL1(TTA)₃** using different recrystallization solvents are similar. It is known that different recrystallization solvent systems can sometimes produce different isomers or packing.¹

1. Nimax, P. R., Reimann, D. & Sünkel, K. Solvent effects on the crystal structure of silver pentacyanocyclopentadienide: supramolecular isomerism and solvent coordination. *Dalton Trans.* **47**, 8476-8482 (2018).

Comment 6: There are mistakes between the Figures and their description in the manuscript, such as Fig. 2b, Fig. 2f, and Fig. 3b.

Response: We thank the reviewer for catching the mistakes. The figures and their description were corrected. Figures and captions throughout the manuscript were checked again.

Comment 7: Full names of **LaL1(TTA)₃**, **LaPhen(TTA)₃**, and **L1** should be added when they first appeared in the text. The structure of **LaPhen(TTA)₃** should also be presented.

Response: Thank you for your suggestion. We have included the full names of **LaL1(TTA)₃**, **LaPhen(TTA)₃** and **L1**. The structure of **LaPhen(TTA)₃** is also presented in the Supplementary Information (Supplementary Fig. 1).

Comments from Reviewer 2:

The manuscript (Ref: NCOMMS-21-25426-T) of the paper entitled “First Reported Charging and Ultralong Phosphorescence of Lanthanide Facilitated Organic Complex”, in my opinion should be considered for publication in the journal: Nature Communications after its minor revision. The article shows very interesting and novel experimental-theoretical studies concerning development of the first ultra-long phosphorescent lanthanide-based complex, operating in the time scale of seconds. The work is very well executed and the performed research are on the top level. However, the following points should be addressed before publication of the article:

Comment 1: Have the luminescence spectra been corrected for the apparatus response? This information should be provided in the characterization section

Response: Yes, the recorded excitation and emission spectra were corrected with correction factors to eliminate the response characteristics. This information was added to the photophysical characteristics subsection.

Comment 2: Figure 1c and d: the Authors may consider presentation of the decay profiles in the exponential representation (Y-axis; intensity scale)

Response: Thank you for your suggestion. The exponential representation of the decay profile (Fig. 1c and 1d in the manuscript) were included in the inset of the respective figure for better understanding.

Comment 3: Figure 3b: maybe a bigger offset/different perspective could be applied to better show the rise (charging) profiles. The most important seem to be the first five profiles, as in the last two ones (20 and 30 s excitation time) we clearly observe the saturation effect, which in fact is not commented by the Authors

Response: Thank you for your suggestion. We have retained the figure because it clearly shows the saturation effect. Figure 3b in the manuscript shows that the luminescence intensity for the last two profiles is due to the completion of the charging rise of **LaL1(TTA)₃**. We have included the comment of the author in the main text for better understanding.

Comment 4: Figure 3c-f: increase the font size in the x-axes (enlarge the numbers)

Response: We thank the reviewer for the careful reading of the manuscript. The font size in the x-axes of Fig. 3c to 3f in the manuscript has been increased for better reading experience.

Comment 5: Figure 7a and b in Extended data: increase the font size in the x-axes (enlarge the numbers)

Response: Again we are grateful for the careful reading of the Reviewer. The font size in the x-axes of Fig. S7a and S7b (now Fig. S9a and S9b) has been increased for better reading experience.

Comment 6: Characterization section: the type (e.g. OPO) and detailed technical parameters (pumping source, pulse duration, energy, etc.) of the pulsed laser used should be provided in the characterization section

Response: Thank you for your suggestion. The type and the detailed parameters of the irradiation source have been added in the photophysical characterization subsection.

Comments from Reviewer 3:

Wong et al reported long phosphorescence of lanthanide complexes, and the 'charging' feature of the system.

Comment 1: Long luminescence from a Ln complexes, or phosphorescent organic chromophore is not new. Long phosphorescence lifetime from an organic chromophore, especially at low temperature, up to seconds, is well known in photochemistry.

Response: Thank you for your suggestion. Despite the fact that some organic chromophores exhibit a long phosphorescence lifetime in the cold, this is not often the case for organo-lanthanide complexes. Taking **GdPhen(TTA)₃** as an example, the phosphorescence lifetime was stated as 2 ms.¹ For a series of La coordinated compounds, the longest phosphorescence reported is 0.29 s.² In this work, the phosphorescence lifetime of **LaL1(TTA)₃** is one order of magnitude longer than previously reported organo-lanthanide complexes. Besides, the long phosphorescence of **LaL1(TTA)₃** can be observed using white light excitation (Supplementary Movie 3), which can lead to different fields of application compared to organic chromophores.

1. e Silva F. R. G., *et al.* Emission quantum yield of europium (III) mixed complexes with thenoyltrifluoroacetate and some aromatic ligands. *J. Alloys Cmpd.* **303-304**, 364-370 (2000).
2. Brinen J. S., Halverson F. & Leto J. R. Photoluminescence of Lanthanide Complexes. IV. Phosphorescence of Lanthanum Compounds. *J. Chem. Phys.* **42**, 4213-4219 (1965).

Comment 2: Some sentences are confusing, such as ‘Intersystem crossing processes from S₁ to T₁ are spin-forbidden but the rates are relaxed upon coordination to a heavy metal, such as for lanthanide complexes’, what does it mean for a ‘rate’ to ‘relax’?

Response: Thank you for your suggestion. We have changed the sentence to “Intersystem crossing from S₁ to T₁ is spin-forbidden but the rate becomes faster upon coordination to a heavy metal, such as a lanthanide ion.”

We have checked through the manuscript thoroughly in order to avoid the inclusion of other confusing sentences.

Comment 3: Figure 4a was used to explain the ‘charging’ feature of the molecules. From a point of view of photochemistry, this is impossible. The T_n state should be with very short lifetime, due to the fast internal conversion (IC) to T₁ state. The mechanism is not convincing.

Response: Thank you for pointing this out. We agree that the decay rate given for T_n was far too low. We have now used a rate constant of 10¹⁰ s⁻¹ in our rate equation model, which is typical for internal conversion.

The rate equation calculations have been repeated using Eqs. (1)-(4), instead of the previously employed Eqs. (2)-(4). The results clearly confirm the experimental finding that the rise (charging) rate constant for T₁ exhibits a linear relation with excitation intensity. Moreover, the “spike” in the charging profile of S₁, which we did not previously understand, is now naturally accounted for by the equilibrium between S₁ and T₁. New figures (Supplementary Figs. 10d-i) have been included to show the results of these calculations, and explanations have been added in the text.

We also included the effects of nonradiative and radiative relaxation separately in the calculations, as well as singlet-singlet excited state absorption. The conclusions are as stated and we avoid unnecessary complication.

REVIEWER COMMENTS

Reviewer #1 (Remarks to the Author):

The authors have carefully addressed all my concerns in this revised manuscript. I recommend its publication as this work reports a timely contribution to the field of phosphorescence.

Reviewer #3 (Remarks to the Author):

The revision is satisfactory, I support the acceptance of the manuscript in its current form.

Response to Reviewers' Comments

We thank the Reviewers for once more reading the manuscript and checking through. It is greatly appreciated.

REVIEWERS' COMMENTS

Reviewer #1 (Remarks to the Author):

The authors have carefully addressed all my concerns in this revised manuscript. I recommend its publication as this work reports a timely contribution to the field of phosphorescence.

Reviewer #3 (Remarks to the Author):

The revision is satisfactory, I support the acceptance of the manuscript in its current form.